# Screening Bacterial Strains Capable of Producing 2,3-Butanediol: Process Optimization and High Diol Production by *Klebsiella oxytoca* FMCC-197

Anastasia Marina Palaiogeorgou, Ermis Ioannis Michail Delopoulos, Apostolis A. Koutinas and Seraphim Papanikolaou *

Department of Food Science and Human Nutrition, Agricultural University of Athens, Iera Odos 75, 11855 Athens, Greece; palaiogeorgou_am@aua.gr (A.M.P.); ermismdelopoulos@outlook.com (E.I.M.D.); akoutinas@aua.gr (A.A.K.)
* Correspondence: spapanik@aua.gr; Tel./Fax: +30-210-529-4700

**Abstract:** In the present investigation, the potential of various newly isolated strains which belong to the Enterobacteriaceae family to produce 2,3-butanediol (BDO), an important bio-based compound, was studied. The most interesting strain, namely *Klebsiella oxytoca* FMCC-197, was selected for further investigation. Commercial (raw) sucrose or molasses, which are important agro-industrial surpluses, were employed as carbon sources for most of the trials performed. Different fermentation parameters (*viz.* incubation te4mperature, utilization of different carbon sources, substrate inhibition, aeration) were tested to optimize the process. Fermentations under non-aseptic conditions were also conducted to investigate the potential of growth of the strain *K. oxytoca* FMCC-197 to surpass the growth of other microorganisms in the culture medium and produce BDO. Besides BDO production, in trials in which molasses was employed as the sole carbon source, significant color removal was observed simultaneously with the production of microbial metabolites. The very high BDO concentration $\approx$115 g L$^{-1}$ was reported in approximately 64 h during a fed-batch bioreactor experiment, using sucrose and molasses as carbon sources at 30 °C, reaching a conversion yield (Y$_{BDO}$) of 0.40 g g$^{-1}$ and a productivity rate (P$_{BDO}$) of 1.80 g L$^{-1}$ h$^{-1}$, while similar results were also obtained at 37 °C. The strain demonstrated remarkable results in non-previously sterilized media, as it produced 58.0 g L$^{-1}$ in 62 h during a fed-batch bioreactor experiment, while the potential to decolorize molasses-based substrates over 40% was also recorded. From the results obtained it is shown that this wild-type strain can be used in large-scale microbial BDO production using various raw materials as fermentative substrates. The wastewater derived after BDO fermentation by *K. oxytoca* FMCC-197 can be disposed relatively safely into the environment.

**Keywords:** 2,3-butanediol; *Klebsiella oxytoca*; molasses; sucrose





## 1. Introduction

The exhaustion of conventional energy resources and their negative environmental impact has led to increasing demands for renewable bioresources and sustainable development. Moreover, byproduct streams from food services, open markets, and households are produced every year in high amounts, so it is recommended that new economically efficient and environmentally friendly management should be applied. Numerous metabolic compounds can be produced via biotechnological methods, including bacterial fermentations on commercial substrates, residues, and low or negative cost materials [1]. One such compound with applications in several industrial fields is that of 2,3-butanediol (BDO), which can be produced via bacterial fermentation. BDO is a colorless and odorless liquid having a very high boiling point (180–184 °C) and low freezing point. Although the fermentation through which BDO is produced was first described in the early part of the 20th century, scientists focused on BDO production in high levels during wartime,

mainly for its feature of being converted to 1,3-butadiene, which is further used in synthetic rubber [2–4]. Additionally, BDO can be applied in the food and pharmaceutical industries, as well as in the manufacturing of printing inks, perfumes, fumigants, moistening and softening agents, explosives, and plasticizers [5,6]. For instance, methyl-ethyl-ketone, the dehydration product of BDO, can be used as an excellent organic solvent for resins and lacquers, while it can also find applications as a liquid fuel having a higher heat of combustion than ethanol [7,8]. BDO can also be dehydrogenated to form acetoin and diacetyl, which are two high added-value compounds. Acetoin is used as an aroma carrier in flavors and essence and its biotechnological production is on high interest recently [9], while diacetyl is important for the organoleptic quality of dairy products, such as cheese, butter, and fermented cream. Furthermore, BDO when mixed with acetone is ketalized producing a "tetramethyl" compound, which is further used as a gasoline blending agent that is similar to the commonly used methyl tert-butyl ether (MTBE) [4,7]. In another case, the dehydration of BDO can also lead to the recovery of 3-buten-2-ol catalyzed by $ZrO_2$ [10].

The microbial species that are predominately referred to in the literature for their ability to produce BDO in significant quantities belong mostly to the bacterial genera *Klebsiella*, *Enterobacter*, *Bacillus*, and *Serratia* [11–13]. Additionally, various strains of the species *Lactobacillus* and *Lactococcus* have the potential to produce BDO in remarkable concentrations [14]. There are three isomers that can be produced via bacterial fermentations, *viz.* dextro-[L-(+)-] and levo-[D-(−)-] forms that are optically active, and an optically inactive meso- form. Different microorganisms produce different isomers, while in general, a mixture of these is produced during bioprocesses [12,14]. Most of the species such as *K. pneumoniae* [15], *K. oxytoca* [16], and *E. aerogenes* [17] produce meso- and dextro- forms. However, *Paenibacillus polymyxa* can produce the pure levo- form that can be used as an antifreeze, due to the special properties of the isomer [18–20].

Although bacterial strains have been used so far for the biotechnological production of BDO, other microorganisms can also ferment various substrates into BDO. Recently, the research has focused on the cultivation of engineered *Saccharomyces cerevisiae* yeast strains on various substrates, producing remarkable final BDO concentrations [21]. For instance, the mutant *S. cerevisiae* SOS4 strain has shown great ability to produce 43.6 g L$^{-1}$ of BDO during a fed-batch fermentation on xylose [22]. In another case, more than 100 g L$^{-1}$ of BDO were synthesized from a mixture of glucose and galactose, which are the two major carbohydrate components that can be found in red algae biomass [23]. As for agricultural wastes, mutants of the yeast species *S. cerevisiae* have been used for the bioconversion of various substrates into BDO, such as in the case of cassava hydrolysates, wherein 132 g L$^{-1}$ of BDO were produced with a volumetric productivity of 1.92 g L$^{-1}$ h$^{-1}$ [24]. It is noted that both hexoses and pentoses were fermented to produce BDO [6]. In bacterial metabolism, pyruvate is initially formed and then it is channeled into a mixture of acetate, lactate, formate, succinate, acetoin, and ethanol, through the mixed-acid–BDO fermentation pathway [12,25]. On the other hand, besides the mentioned compounds, remarkable BDO production has been noted when glycerol, the main side product derived from biodiesel facilities, was added into the culture medium as a carbon source [13].

In the present study, molasses has been tested as a principal or supplementary fermentation substrate in batch and fed-batch experiments. Molasses, the liquid byproduct of sugar cane or sugar beet processing into sugar, besides sugars (44–60% *w/w*), contain a high-molecular-weight dark-brown pigment called melanoidin. Molasses-derived wastewaters contain a heavy organic load and include high concentrations of nutrients and minerals such as nitrogen, phosphorus, and potassium, which may lead to eutrophication phenomena. They are also characterized by a low pH and strong odor and dark color that can affect marine life. For these reasons, the research so far has focused not only on the bioconversion of molasses on high value-added compounds but also on its color removal [26,27]. For instance, several studies have highlighted the production of bioethanol by *Saccharomyces cerevisiae* via the alcoholic fermentation of molasses [28–30]. Other studies have focused

on the acetic acid fermentation of soybean molasses [31] and lactic acid production from molasses using several bacterial species [32,33].

The process for the microbial production of BDO is reportedly influenced by various fermentation parameters. The main factors that affect the final BDO production are pH, incubation temperature, aeration, agitation, and substrate composition. The appropriate inoculum preparation can also influence the final product concentration. In this study, a screening using several newly isolated and wild-type bacterial strains was initially carried out under anaerobic conditions, to investigate the potential of microorganisms to assimilate and convert glucose and sucrose into BDO. Among the strains that were cultivated, the strain *K. oxytoca* FMCC-197 has shown the most promising results concerning the bioconversion yield of BDO on sucrose consumed ($Y_{BDO}$) and a productivity rate ($P_{BDO}$), and it was selected for further investigation in batch and fed-batch experiments. Different parameters (carbon source, initial concentration, temperature, trials under aerobiosis or anaerobiosis, previous pasteurization or sterilization of the medium) were investigated in batch experiments. Taking into consideration the most promising results obtained from the previous experiments, fed-batch bioreactor fermentations applying the preferable conditions were carried out using molasses as the sole carbon source or along with sucrose in different aeration strategies. Finally, the decolorization of molasses was also assessed. Biochemical and technological considerations of the process were considered. The results obtained from the current research indicate that the strain *K. oxytoca* FMCC-197 is a competitive BDO producer and its application in biotechnological processes can contribute to the development of alternative ways for efficient and environmentally friendly waste management.

## 2. Materials and Methods

Microorganisms. For the initial screening process, eight newly isolated, food-derived strains (*Enterobacter ludwigii* FMCC-204, *E. aerogenes* FMCC-9, *E. aerogenes* FMCC-10, *Citrobacter freundii* FMCC-207, *Klebsiella oxytoca* FMCC-197, *C. freundii* FMCC-8, *C. farmeri* FMCC-5, and *C. farmeri* FMCC-7) that belong to the Enterobacteriaceae family were assessed for their ability to consume sugars (i.e., analytical-grade glucose and commercial-type sucrose) and produce BDO under anaerobic conditions in batch-type fermentations carried out in 1-L Duran bottles filled with 800 mL of medium. These bottles were used instead of the simpler serum bottles, since even in the "screening" experiment it was desirable to perform full kinetics, and, therefore, many experimental points needed to be taken into account. The strains were isolated from foodstuffs, were identified and characterized at the Department of Food Science and Technology, and were deposited in the culture collection of this Department [34–37]. All strains were preserved at $-80\ ^{\circ}$C in Tryptic Soy Broth, supplemented with 20% glycerol from Sigma Chemical, St. Louis, MO, USA. Prior to each experimental application, each strain was cultivated in Tryptic Soy Broth at the optimal temperature for 24 h. Subsequently, petri dishes were inoculated with this culture and incubated at T = 30 $^{\circ}$C for 24 h. This culture was then used for the preparation of pre-cultures [4].

Culture conditions. *i. Anaerobic batch fermentations in Duran bottles.* A preliminary assessment of glucose and sucrose consumption and BDO production from various strains was conducted in 1-L Duran bottles under anaerobic conditions. The fermentations were performed on a synthetic (MRS) medium containing (per L of medium): peptone, 5 g; meat extract, 5 g; yeast extract, 2.5 g; $K_2HPO_4$, 2 g; $CH_3COONa$, 5 g; $MgSO_4$, 0.41 g. Commercial sucrose purchased from a grocery market and sugarcane molasses, provided by the sugarcane industry Cruz Alta (Guarani, Sao Paulo, Brazil), (containing (in % $w/w$) sucrose 43.5%, glucose 7.3%, fructose 6.3%, protein (expressed as total Kjeldahl nitrogen $\times$ 6.25) 3.2%, moisture 29.7%, solids 70.4%) were used as carbon sources. The results of the element composition of sugarcane molasses, as it was analyzed and determined in the laboratory, are presented in Table 1. During the anaerobic experiments, the MRS broth was supplemented with $\approx$30 g $L^{-1}$ of sucrose. Further experiments using the most interesting

amongst the tested strains, namely the strain *K. oxytoca* FMCC-197, to produce BDO under anaerobic conditions were also conducted in 1-L Duran bottles, using molasses and different initial concentrations of sucrose in order to examine the substrate inhibition in growth. The anaerobic fermentations were conducted in a working volume of 80 mL and inoculated with 10% (*v/v*) of a pre-culture that had been prepared using MRS media supplemented with 10 g $L^{-1}$ of analytical-grade glucose. Anaerobic conditions were achieved during the batch experiments in Duran bottles as the medium was sparged with $N_2$ for 30 min before autoclaving. The initial pH value was $7.0 \pm 0.2$ and no control was applied until the end of the fermentation. The Duran bottles were incubated in the orbital shaker at an agitation speed of 180 rpm at T = 30 °C. Samples were taken every 2 to 3 h depending on the experiment.

**Table 1.** Element composition of sugarcane molasses employed as a carbon source throughout the current study.

| Elements | ppm |
|:---:|:---:|
| Ca | 6.67 |
| S | 5.23 |
| Mg | 4.70 |
| K | 1.75 |
| P | 0.21 |
| Fe | 0.12 |
| Mn | 0.03 |
| Zn | 0.003 |
| Co | <0.0005 |

*ii. Aerobic batch fermentations in shake flasks.* The substrate inhibition during cultivation of the strain *K. oxytoca* FMCC-197 was also examined during aerobic fermentations in 500-mL non-baffled shake flasks using various initial sucrose concentrations. In addition, batch fermentations in shake flasks were conducted using various carbon sources (glucose, fructose, mannose, xylose, arabinose, galactose, and molasses) at an initial concentration of sugar adjusted to ≈30 g $L^{-1}$. All the carbon sources that were used during the fermentations were of analytical grade (purity ≈ 99.5%). All the previous experiments were conducted at T = 30 °C. The temperature effect on bacterial growth was also investigated through aerobic fermentations. Thus, 6 different temperatures (i.e., T = 25, 30, 34, 37, 40, and 42 °C) were applied during the experiments. To ensure that the selected temperature value was stable during the whole process, a thermometer was placed in the incubator and in all cases, there was no significant variation. Bacterial growth of *K. oxytoca* FMCC-197 was studied using a wide range of initial sucrose concentrations to identify the maximum value in which there is no substrate inhibition. Two sets of batch experiments were conducted under different aeration modes. Ten different initial sucrose concentrations (*viz.* ≈5, ≈10, ≈15, ≈20, ≈30, ≈60, ≈90, ≈110, ≈120, ≈150 g $L^{-1}$) were used during cultivations under anaerobic conditions (Duran bottles) and thirteen different initial sucrose concentrations (*viz.* ≈5, ≈10, ≈15, ≈20, ≈40, ≈60, ≈80, ≈90, ≈110, ≈120, ≈130, ≈150, ≈160 g $L^{-1}$) were used under aerobic conditions (shake flask trials) at T = 30 °C. Samples were taken every 30 min and the $\mu_{max}$ value of each of the fermentations carried out was assessed. The specific growth rate of each sample was calculated by fitting the equation $\ln\left(\frac{X}{X_0}\right) = f(t)$. on the experimental data within the early exponential growth phase (X is the dry cell weight (DCW) concentration in g $L^{-1}$). For each case, $\mu_{max}$ was the slope of the trendline that occurred. All shake flask cultures were conducted in a working volume of 100 mL and inoculated with 10% (*v/v*) pre-culture. The initial pH value was $7.0 \pm 0.2$ and no fixation occurred until the end of the fermentation. The incubation of shake flasks took place in the orbital shaker at an agitation speed of 180 rpm. Samples were taken every 2 to 3 h depending on the experiment.

*iii. Fed-batch bioreactor fermentations.* Taking into consideration the results obtained from the batch experiments, the study further focused on the optimization of BDO production through fed-batch experiments in a 2-L bioreactor (Infors HT, Type Labfors, Basel, Switzerland). Molasses was used as the initial carbon source and when the carbon source concentration was low, pulses of a concentrated sucrose solution (600 g $L^{-1}$) also containing 5% (*w/v*) yeast extract or a concentrated molasses solution ($\approx$100 g $L^{-1}$ of total sugars) were added into the culture (therefore, fed-batch cultures with intermittent feeding were performed). The substrate was supplemented with MRS media. The fed-batch cultures were conducted at T = 30 °C under anaerobic and aerobic conditions, starting with 45 g $L^{-1}$ and 80 g $L^{-1}$ for the total sugars concentration. In order to achieve anaerobic conditions, the medium was sparged with $N_2$ for 20 min before sterilization. In the case of aerobic conditions, 1 vvm of aeration was provided into the culture medium while the dissolved oxygen tension (DOT) was constantly maintained at values $\geq$ 20% *v/v*, achieved with a cascade agitation rate from 180 rpm to 400 rpm. The initial pH was adjusted to the value 7.0 $\pm$ 0.2 before autoclaving and was controlled by the automatic addition of 5 M NaOH when the value was under 6.0.

*iv. Fed-batch fermentations in shake flasks.* Aiming to further assess the potential of the process, a fed-batch experiment was conducted in a 2-L shake flask presenting 500 mL of working volume. Sucrose was used as the sole carbon source during the whole fermentation at an initial concentration of 80 g $L^{-1}$ and pulses of a concentrated sucrose solution (600 g $L^{-1}$) also containing 5% (*w/v*) yeast extract were added into the culture when the substrate concentration was low. In this type of experiment, trials were performed under aseptic conditions (*viz.* media that were previously sterilized) and non-aseptic conditions. For the non-aseptic conditions, the culture medium was thermally treated to T = 80 °C for 15 min (high pasteurization therefore occurred) and the inoculum was 15% (*v/v*) of the final working volume. The agitation speed was 180 rpm and the pH was adjusted to the value 7.0 before autoclaving and was controlled by the automatic addition of 5 M NaOH when the value was lower than 6.0.

Analytical methods. Cell concentration (X, g $L^{-1}$) was determined through dry cell weight analysis (wet biomass was put at T = 90 $\pm$ 5 °C until constant weight). Cells were collected by centrifugation (9000$\times$ *g*/15 min, T = 9 °C) in a Hettich Universal 320-R (Hettich Zentrifugen, Tuttlingen, Germany) centrifuge and washed twice with distilled water. The pH value was measured with a selective pH meter (Jenway, Jenway 3020, Staffordshire, UK). Concentrations of monosaccharides, BDO, and organic acids were determined with High Performance Liquid Chromatography (HPLC) analysis equipped with a Shimadzu RI detector and an Aminex HPX-87H (300 mm $\times$ 7.8 mm, BioRad, Hercules, CA, USA) column. Operating conditions were as follows: sample volume 20 $\mu$L; mobile phase 0.010 M $H_2SO_4$; flow rate 0.6 mL $min^{-1}$; column temperature T = 45 °C. The substrate containing and/or sucrose was hydrolyzed to glucose and fructose prior to HPLC analysis. This was achieved by mixing 100 $\mu$L of 10% (*v/v*) $H_2SO_4$ solution with 500 $\mu$L of supernatant followed by heating the mixture at T = 100 °C for 30 min [38]. The concentration of sucrose or carbohydrates found in the molasses is expressed as total sugars, including the concentration of both glucose and fructose, while the dilution of the supernatant mixed with the acid was taken into consideration. The bioconversion yield of BDO produced on sucrose consumed was assessed via the equation $Y_{BDO}$ = g BDO produced/g sucrose consumed, while the volumetric productivity rate of BDO produced was assessed via the equation $P_{BDO}$ = (g BDO produced)/(L fermentation volume $\times$ fermentation time in hours).

Medium decolorization containing molasses was assessed by measuring the decrease in absorbance at 475 nm (U-2000, Hitachi, Tokyo, Japan) as described in previous studies [39,40]. The absorbance (475 nm) of the fermentation medium (diluted 10 times) before inoculation was determined as the set-point. The difference between the absorbance of the set-point and each experimental point was expressed in %. All data presented are the average of two independent experiments carried out under the same culture conditions.

The determination of DOT (%, $v/v$) in the shake flask cultures was performed using a Lonibond Sensodirect OXI 200 oxygen meter (Lonibond GmbH, Berlin, Germany), as explained explicitly in [41,42]. Precisely, before harvesting, the shaker in which the shake flask fermentations were performed was switched off and the oxygen measuring probe was placed into the flask. Then, the shaker was again switched on and the measurement was taken after DOT equilibration (usually within the next 10 min after the shaker was again switched on). The measurement of DOT was represented by both oxygen saturation (% $v/v$) and dissolved oxygen concentration into the liquid medium (in mg $L^{-1}$). For the assessment of the specific oxygen consumption rate ($q_{O2}$, in mg $mg^{-1}$ $h^{-1}$), after DOT equilibration the shaker was again switched off and the oxygen concentration (in mg $L^{-1}$) was recorded every 5 s. The slope of the curve $[O_2] = f(t)$ represented the oxygen consumption rate (mg $L^{-1}$ $h^{-1}$), and the value of $q_{O2}$ was found after division of the oxygen consumption rate by the respective DCW value, corresponding to the measured flask [42].

## 3. Results

Initial trials for sucrose assimilation and BDO production. Eight newly isolated food-derived strains (*viz. Enterobacter ludwigii* FMCC-204, *E. aerogenes* FMCC-9, *E. aerogenes* FMCC-10, *Citrobacter freundii* FMCC-207, *C. freundii* FMCC-8, *C. farmeri* FMCC-5, *C. farmeri* FMCC-7, and *Klebsiella oxytoca* FMCC-197) were evaluated for their ability to grow and efficiently consume sugars (*viz.* glucose and commercial-type sucrose), leading to the production of BDO. The strains that were cultivated along with the total substrate consumption, the final biomass, and BDO production are presented in Table 2. As shown, five amongst the tested strains could efficiently consume glucose and sucrose, producing BDO in highly promising yields. For instance, in all cases the bioconversion yield of BDO on sucrose consumed ($Y_{BDO}$) was $\geq 0.40$ g $g^{-1}$, while the volumetric productivity rates $P_{BDO}$ (in g $L^{-1}$ $h^{-1}$) varied, depending on the strain. The strain *K. oxytoca* was selected for further investigation as it was the one that combined significantly high productivity rates both in sucrose (0.72 g $L^{-1}$ $h^{-1}$) and in glucose (0.88 g $L^{-1}$ $h^{-1}$), along with a remarkable yield of 0.41 g $g^{-1}$ and 0.48 g $g^{-1}$, respectively (see Table 2). In contrast, three of the tested strains (belonging to the species *C. freundii* and *C. farmeri*) could not assimilate sucrose and the biomass accumulation was slight after 20 h of batch cultivation, while growth on glucose was not accompanied by the production of BDO, but only of acetic acid.

**Table 2.** Selection of wild-type enteric group bacterial strains growing on sugars (glucose or sucrose) under anaerobic conditions in 1-L Duran bottles. Representation of maximum biomass production ($X_{max}$), substrate (sugar) consumption, final BDO concentration, conversion yield on sugars consumed ($Y_{BDO}$), and productivity ($P_{PDO}$) in batch fermentations. Culture conditions: T = 30 °C; agitation rate = 180 rpm; initial pH = 7.0. Each point is the mean value of two independent measurements.

| Strain | Carbon Source | Substrate Consumed (g $L^{-1}$) | $X_{max}$ (g $L^{-1}$) | BDO (g $L^{-1}$) | $Y_{BDO}$ (g $g^{-1}$) | $P_{BDO}$ (g $L^{-1}$ $h^{-1}$) | Fermentation Time (h) |
|---|---|---|---|---|---|---|---|
| *Enterobacter ludwigii* | Sucrose | 29.2 ± 0.4 | 2.5 ± 0.2 | 12.5 ± 0.3 | 0.43 ± 0.01 | 0.63 ± 0.01 | 20 |
| FMCC-204 | Glucose | 27.9 ± 0.3 | 3.2 ± 0.1 | 13.3 ± 0.2 | 0.48 ± 0.02 | 0.72 ± 0.02 | 18 |
| *Enterobacter aerogenes* | Sucrose | 30.0 ± 1.0 | 3.1 ± 0.1 | 13.0 ± 0.5 | 0.43 ± 0.01 | 0.65 ± 0.03 | 20 |
| FMCC-9 | Glucose | 29.3 ± 0.2 | 3.3 ± 0.1 | 12.7 ± 0.2 | 0.43 ± 0.01 | 0.71 ± 0.01 | 18 |
| *Enterobacter aerogenes* | Sucrose | 29.8 ± 1.1 | 3.3 ± 0.5 | 12.0 ± 1.0 | 0.40 ± 0.02 | 0.60 ± 0.05 | 20 |
| FMCC-10 | Glucose | 31.1 ± 1.2 | 2.9 ± 0.2 | 14.2 ± 1.2 | 0.46 ± 0.03 | 0.79 ± 0.02 | 18 |
| *Citrobacter freundii* | Sucrose | 28.7 ± 2.2 | 2.5 ± 0.0 | 12.5 ± 0.3 | 0.44 ± 0.04 | 0.50 ± 0.01 | 25 |
| FMCC-207 | Glucose | 29.8 ± 1.2 | 3.5 ± 0.2 | 12.3 ± 0.3 | 0.41 ± 0.02 | 0.68 ± 0.02 | 18 |
| *Klebsiella oxytoca* | Sucrose | 31.5 ± 0.8 | 2.1 ± 0.3 | 13.0 ± 0.8 | 0.41 ± 0.02 | 0.72 ± 0.03 | 18 |
| FMCC-197 | Glucose | 29.1 ± 1.6 | 3.9 ± 1.0 | 14.0 ± 0.5 | 0.48 ± 0.02 | 0.88 ± 0.04 | 16 |
| *Citrobacter freundii* | Sucrose | - | - | - | - | - | 24 |
| FMCC-8 | Glucose * | 25.3 ± 0.5 | 3.5 ± 0.2 | - | - | - | 24 |

**Table 2.** *Cont.*

| Strain | Carbon Source | Substrate Consumed (g L$^{-1}$) | X$_{max}$ (g L$^{-1}$) | BDO (g L$^{-1}$) | Y$_{BDO}$ (g g$^{-1}$) | P$_{BDO}$ (g L$^{-1}$ h$^{-1}$) | Fermentation Time (h) |
|---|---|---|---|---|---|---|---|
| *Citrobacter farmeri* FMCC-5 | Sucrose | - | - | - | - | - | 24 |
| | Glucose * | 24.2 ± 0.2 | 2.5 ± 0.2 | - | - | - | 24 |
| *Citrobacter farmeri* FMCC-7 | Sucrose | - | - | - | - | - | 24 |
| | Glucose * | 26.3 ± 0.5 | 3.0 ± 0.2 | - | - | - | 24 |

* At the end of the fermentation acetic acid was produced at a final concentration $\leq 7.0$ g L$^{-1}$.

Initial trials for sucrose assimilation and BDO production using the strain *K. oxytoca* FMCC-197. Given that sucrose proved to be an efficient substrate for *K. oxytoca*, shake flask cultures with $\approx$30 g L$^{-1}$ as the initial concentration of sugars were performed to investigate the ability of the strain to assimilate the substrate and produce BDO under aerobic conditions. The results obtained were compared to those derived from the relevant anaerobic trials, while experiments with molasses (this food-derived byproduct contained mainly sucrose as the principal carbon source; see Section 2, "Materials and Methods") were also performed under aerobic and anaerobic conditions. As shown in Table 3, the whole substrate was successfully consumed at 10 h of the shake flask experiments and the productivity rate was remarkably increased, reaching a value of 1.40 g L$^{-1}$ h$^{-1}$, as compared to the anaerobic experiment. Similar results were also recorded with the utilization of molasses (see Table 3). These experiments suggest that the strain *K. oxytoca* FMCC-197 can consume sucrose both under anaerobic and aerobic conditions, however, the presence of O$_2$ in the culture medium seems to lead to a more efficient substrate assimilation, decreasing the duration of the fermentation, which is in some disagreement with the literature reports that indicate the necessity for the application of anaerobic/microaerophilic conditions in the growth environment in order to secure significant BDO production by most of the bacterial strains implicated in this bioprocess [14]. In fact, increasing the O$_2$ supply rate leads to higher biomass production, which can explain higher productivity rates.

**Table 3.** Cultures of *K. oxytoca* FMCC-197 in Duran bottles under anaerobic conditions and in shake flasks under aerobic conditions using commercial sucrose and molasses as carbon sources at initial sugar concentrations of $\approx$30 g L$^{-1}$. Representation of maximum biomass production (X$_{max}$), substrate consumption, final BDO concentration, conversion yield on sugars consumed (Y$_{BDO}$), and productivity (P$_{PDO}$) in batch fermentations. Culture conditions: T = 30 °C; agitation rate = 180 rpm; initial pH = 7.0. Each point is the mean value of two independent measurements.

| | Cultivation Mode | Initial Sugar Concentration (g L$^{-1}$) | Total Sugars Consumed (g L$^{-1}$) | X$_{max}$ (g L$^{-1}$) | BDO (g L$^{-1}$) | Y$_{BDO}$ (g g$^{-1}$) | P$_{BDO}$ (g L$^{-1}$ h$^{-1}$) | Fermentation Time (h) |
|---|---|---|---|---|---|---|---|---|
| Sucrose | Anaerobic cultures | 31.5 ± 0.8 | 31.5 ± 0.8 | 2.1 ± 0.3 | 13.0 ± 0.8 | 0.41 ± 0.02 | 0.72 ± 0.03 | 18 |
| | Aerobic cultures | 37.1 ± 0.1 | 37.1 ± 0.1 | 6.4 ± 0.2 | 14.0 ± 0.1 | 0.38 ± 0.01 | 1.40 ± 0.01 | 10 |
| Molasses | Anaerobic cultures | 31.2 ± 0.5 | 31.2 ± 0.5 | 3.2 ± 0.2 | 12.0 ± 0.2 | 0.38 ± 0.02 | 0.6 ± 0.01 | 20 |
| | Aerobic cultures | 35.2 ± 1.5 | 35.2 ± 1.5 | 5.9 ± 0.1 | 14.1 ± 0.2 | 0.40 ± 0.01 | 1.41 ± 0.01 | 10 |

Evaluation of the ability of the strain to assimilate different carbon sources. The present study further focused on the evaluation of the ability of the strain to convert various carbohydrates into BDO. Eight different individual carbon sources (*viz.* sucrose, glucose, fructose, mannose, xylose, arabinose, galactose, and molasses) were added into the culture medium during shake flask experiments at the incubation temperature T = 30 °C. The substrate consumption along with the biomass, BDO, and other organic acids production are presented in Table 4. The strain *K. oxytoca* FMCC-197 has shown great potential in the assimilation of all the carbon sources that were used. Except for xylose, in all other cases

the conversion yield $Y_{BDO}$ ranged between 0.41 and 0.47 g g$^{-1}$ (values ranging between 82% and 94% of the maximum theoretical yield of the conversion). As for the volumetric productivity, the maximum values (~1.00 g L$^{-1}$ h$^{-1}$) were obtained when glucose, fructose, galactose, sucrose, and molasses were added as carbon sources into the culture medium. It should be stressed that the fermentation duration depends on the carbon source used. As shown, xylose was assimilated at the lowest rate (22 h) while the substrate consumption in all other carbon sources varied from 10 to 14 h. Significant biomass production and noticeable volumetric productivity of synthesized BDO were recorded when sucrose was employed as the sole carbon source, and therefore this substrate was maintained for the forthcoming trials. Moreover, it should also be stressed that in the current study, in spite of the fact that aerobic conditions were set (i.e., cultures in shake-flask mode), metabolites that are practically produced under anaerobic conditions were produced. This is indeed somehow paradoxical and surprising, given that the fermentation of BDO is practically mostly an anaerobic/microaerophilic process. Nevertheless, in full agreement with many reports that have appeared in the literature [5,6,11–14], the current investigation provides sufficient evidence for the biosynthesis and the subsequent remarkable production of BDO under full aerobic conditions, suggesting that a part of the NADH$_2$ co-enzymes that had been produced through the glycolysis pathway are recycled through the sugar pathway that yields → BDO, while simultaneously another part of the intracellular pool of these NADH$_2$ co-enzymes is recycled through the oxidative phosphorylation pathway, indicating simultaneous oxidative and fermentative features realized under fully aerobic conditions in the presently used strain.

**Table 4.** Maximum biomass production ($X_{max}$), substrate (sugar) consumption, final BDO concentration, other organic compounds' produced concentrations, conversion yield ($Y_{BDO}$), and productivity ($P_{BDO}$) in batch fermentations using different carbon sources under aerobic conditions for *K. oxytoca* growing in shake flask experiments. Culture conditions: T = 30 °C; agitation rate = 180 rpm; initial pH = 7.0. Each point is the mean value of two independent measurements.

| Carbon Source | Total Sugars Consumed (g L$^{-1}$) | $X_{max}$ (g L$^{-1}$) | BDO (g L$^{-1}$) | Ethanol (g L$^{-1}$) | Succinic (g L$^{-1}$) | Lactic (g L$^{-1}$) | $Y_{BDO}$ (g g$^{-1}$) | $P_{BDO}$ (g L$^{-1}$ h$^{-1}$) | Fermentation Time * (h) |
|---|---|---|---|---|---|---|---|---|---|
| Glucose | 27.7 ± 1.1 | 6.6 ± 0.4 | 10.9 ± 0.2 | 2.3 ± 0.3 | 2.4 ± 0.1 | 0.4 ± 0.1 | 0.45 ± 0.01 | 1.03 ± 0.02 | 12 |
| Fructose | 28.3 ± 0.9 | 5.8 ± 0.1 | 11.9 ± 1.2 | 2.3 ± 0.1 | 2.5 ± 0.1 | 0.5 ± 0.1 | 0.42 ± 0.01 | 0.99 ± 0.10 | 12 |
| Mannose | 23.8 ± 1.1 | 5.8 ± 0.2 | 10.7 ± 0.2 | 2.2 ± 0.2 | 2.2 ± 0.1 | 0.5 ± 0.1 | 0.45 ± 0.03 | 0.82 ± 0.02 | 13 |
| Xylose | 16.0 ± 0.7 | 6.0 ± 0.1 | 4.4 ± 0.2 | 0.0 ± 0.0 | 0.8 ± 0.1 | 0.0 ± 0.0 | 0.30 ± 0.01 | 0.22 ± 0.01 | 22 |
| Arabinose | 17.4 ± 1.5 | 7.0 ± 0.4 | 8.2 ± 0.2 | 0.8 ± 0.1 | 1.1 ± 0.3 | 0.0 ± 0.0 | 0.47 ± 0.02 | 0.58 ± 0.01 | 14 |
| Galactose | 28.5 ± 0.7 | 6.8 ± 0.0 | 11.8 ± 0.3 | 2.1 ± 0.4 | 1.5 ± 0.4 | 0.0 ± 0.0 | 0.41 ± 0.01 | 0.98 ± 0.02 | 12 |
| Sucrose | 37.1 ± 0.1 | 6.4 ± 0.2 | 14.0 ± 0.1 | 1.3 ± 0.1 | 1.6 ± 0.1 | 0.2 ± 0.0 | 0.38 ± 0.01 | 1.40 ± 0.01 | 10 |
| Molasses | 35.2 ± 1.5 | 5.9 ± 0.1 | 14.1 ± 0.2 | 1.1 ± 0.1 | 1.9 ± 0.3 | 0.2 ± 0.0 | 0.40 ± 0.01 | 1.41 ± 0.01 | 10 |

* Fermentations were extended after the indicated time and no more sugar had been consumed after the given fermentation point.

Impact of incubation temperature on growth and products formation. Six different incubation temperature values (i.e., T = 25, 30, 34, 37, 40, and 42 °C) were applied to shake flask cultivations, in order to examine the effect of temperature on the final biomass and BDO accumulation. Sucrose was used as the carbon source in the culture medium, in an initial concentration of *c.* 30–40 g L$^{-1}$. Table 5 summarizes the total substrate assimilation, the biomass production, and the BDO and other byproducts' biosynthesis at the end of each experiment, using the different mentioned incubation temperature values. From the results obtained it is obvious that BDO production along with the fermentation duration are related to the temperature applied on the culture medium. In fact, the bioconversion yield $Y_{BDO}$ ranged from 0.38 g g$^{-1}$ to 0.45 g g$^{-1}$ in values of temperature ranging between 25 °C and 34 °C, while at higher temperatures the yields obtained were dramatically decreased. As for the volumetric productivity $P_{BDO}$, in temperatures from 30 °C to 37 °C the highest values were noted, which corresponded to 1.15 g L$^{-1}$ h$^{-1}$ to 1.40 g L$^{-1}$ h$^{-1}$. These findings

indicate that temperature values higher than 37 °C eliminate the enzyme activity of *K. oxytoca* and BDO production is reduced dramatically. The incubation temperature T = 30 °C was the most promising and it was maintained for the experiments that followed.

**Table 5.** Maximum biomass production ($X_{max}$), substrate consumption, final BDO and other organic acid concentration, conversion yield ($Y_{BDO}$), and productivity ($P_{BDO}$) in batch fermentations using commercial sucrose as a carbon source at $\approx$30–40 g L$^{-1}$ initial concentration in different temperatures under aerobic conditions. The accuracy of the temperature was measured by a thermometer placed in the incubator during the whole process. Culture conditions: agitation rate = 180 rpm; initial pH = 7.0. Each point is the mean value of two independent measurements.

| Temperature °C | Total Sugars Consumed (g L$^{-1}$) | $X_{max}$ (g L$^{-1}$) | BDO (g L$^{-1}$) | Ethanol (g L$^{-1}$) | Succinic (g L$^{-1}$) | Lactic (g L$^{-1}$) | $Y_{BDO}$ (g L$^{-1}$) | $P_{BDO}$ (g L$^{-1}$) | Fermentation Time (h) |
|---|---|---|---|---|---|---|---|---|---|
| 25 | 37.5 ± 0.7 | 4.9 ± 0.7 | 16.8 ± 1.3 | 1.4 ± 0.3 | 1.4 ± 0.1 | 0.2 ± 0.0 | 0.45 ± 0.03 | 0.93 ± 0.07 | 18 |
| 30 | 37.1 ± 0.1 | 6.4 ± 0.2 | 14.0 ± 0.1 | 1.3 ± 0.1 | 1.6 ± 0.1 | 0.2 ± 0.1 | 0.38 ± 0.00 | 1.40 ± 0.01 | 10 |
| 34 | 33.9 ± 2.7 | 4.8 ± 0.4 | 14.4 ± 0.9 | 1.1 ± 0.1 | 1.5 ± 0.3 | 0.1 ± 0.0 | 0.42 ± 0.01 | 1.20 ± 0.08 | 12 |
| 37 | 39.8 ± 1.6 | 4.8 ± 0.4 | 12.6 ± 0.6 | 2.5 ± 0.0 | 2.1 ± 0.1 | 0.7 ± 0.1 | 0.32 ± 0.02 | 1.15 ± 0.01 | 11 |
| 40 | 40.0 ± 0.4 | 5.1 ± 0.2 | 13.8 ± 0.6 | 1.5 ± 0.0 | 1.9 ± 0.1 | 0.3 ± 0.1 | 0.35 ± 0.02 | 0.86 ± 0.07 | 16 |
| 42 | 26.4 ± 1.4 | 7.2 ± 0.8 | 8.0 ± 0.9 | 0.5 ± 0.1 | 0.0 ± 0.0 | 0.0 ± 0.0 | 0.30 ± 0.02 | 0.31 ± 0.03 | 26 |

Impact of different initial sucrose concentrations upon the microbial growth of *K. oxytoca* FMCC-197 and BDO formation during batch experiments. Given that sucrose proved to be an efficient substrate for *K. oxytoca,* different initial sucrose quantities were employed, to investigate the ability of the strain to assimilate various initial concentrations of this sugar under anaerobic and aerobic conditions. Two sets of batch experiments were conducted under different aeration modes. In particular, ten different initial sucrose concentrations (*viz.* $\approx$5, $\approx$10, $\approx$15, $\approx$20, $\approx$30, $\approx$60, $\approx$90, $\approx$110, $\approx$120, $\approx$150 g L$^{-1}$) were used during cultivations under anaerobic conditions (Duran bottles) and thirteen different initial sucrose concentrations (*viz.* $\approx$5, $\approx$10, $\approx$15, $\approx$20, $\approx$40, $\approx$60, $\approx$80, $\approx$90, $\approx$110, $\approx$120, $\approx$130, $\approx$150, $\approx$160 g L$^{-1}$) were used during cultivations under aerobic (shake flasks) conditions. This wide range of initial substrate concentrations was chosen in order to specifically note the threshold value of substrate inhibition. Therefore, variations from indeed low initial sugar concentrations (i.e., $\approx$5 g L$^{-1}$) to very high ones (i.e., $\geq$120 g L$^{-1}$) would demonstrate the potential inhibition at the high initial sugar concentrations, as well as to which initial substrate concentration the $\mu_{max}$ value would attain its highest value. The growth, the lag phase duration, and the maximum specific growth rate related to each initial sucrose concentration are presented in Tables 6 and 7, respectively. From the results obtained it is proven that the strain *K. oxytoca* FMCC-197 was able to successfully grow in a wide range of substrate concentrations without inhibition under both aeration modes. Specifically, during the anaerobic conditions, the strain was unable to assimilate the whole substrate when sucrose quantities were >90 g L$^{-1}$. In high initial sucrose concentrations, part of the substrate remained unconsumed and the specific growth rate reached its highest value ($\mu_{max}$ > 0.50 h$^{-1}$) at 30 g L$^{-1}$ of the initial substrate concentration. Biomass production was favored by the aeration, since noticeably higher DCW production and, mainly, higher conversion yield $Y_{X/S}$ values occurred throughout the shake flask experiments as compared to the anaerobic Duran bottle experiments (see Tables 6 and 7). At the end of the previous fermentations, a slight formation of other organic acids was observed, and their concentrations are not noted in the relevant tables.

**Table 6.** Growth of *K. oxytoca* FMCC-197 under anaerobic conditions in Duran bottles using various initial sucrose concentrations employed. Representation of the maximum specific growth rate ($\mu_{max}$) of the strain, maximum biomass production ($X_{max}$), yield of biomass produced per substrate consumed ($Y_{X/S}$ in g g$^{-1}$) in Duran bottle fermentations carried out at T = 30 °C. Culture conditions: growth in 1-L Duran bottles filled with 800 mL; agitation rate = 180 rpm; initial pH = 7.0. Each point is the mean value of two independent measurements.

| Initial Sucrose Concentration (g L$^{-1}$) | $\mu_{max}$ (h$^{-1}$) | Sugar Consumed (g L$^{-1}$) | $X_{max}$ (g L$^{-1}$) | $Y_{X/S}$ (g g$^{-1}$) | BDO (g L$^{-1}$) | Lag Phase Duration (h) | Fermentation Time (h) |
|---|---|---|---|---|---|---|---|
| ≈5 | 0.40 | 5.5 | 0.5 | 0.09 | 2.1 | 1.0 | 7.5 |
| ≈10 | 0.42 | 10.5 | 1.1 | 0.10 | 3.5 | 1.0 | 10 |
| ≈15 | 0.44 | 14.5 | 1.5 | 0.10 | 4.5 | 1.5 | 12 |
| ≈20 | 0.45 | 19.0 | 2.0 | 0.11 | 6.0 | 1.5 | 15 |
| ≈30 | 0.50 | 31.5 | 2.1 | 0.07 | 13.0 | 2.5 | 18 |
| ≈60 | 0.40 | 62.2 | 3.8 | 0.06 | 27.5 | 3.0 | 24 |
| ≈90 | 0.35 | 83.0 * | 4.3 | 0.05 | 31.8 | 4.0 | 28 * |
| ≈110 | 0.20 | 87.2 ** | 6.2 | 0.07 | 38.8 | 2.5 | 32 ** |
| ≈120 | 0.10 | 95.3 *** | 4.9 | 0.05 | 35.5 | 5.0 | 34 *** |
| ≈150 | 0.06 | 101.3 **** | 5.4 | 0.05 | 35.6 | 6.0 | 48 **** |

* Initial sugars at 92.0 g L$^{-1}$, fermentation was extended after 28 h and no more sugar was consumed; ** initial sugars at 108 g L$^{-1}$, fermentation was extended after 32 h and no more sugar was consumed; *** initial sugars at 124.2 g L$^{-1}$, fermentation was extended after 34 h and no more sugar was consumed; **** initial sugars at 149.8 g L$^{-1}$, fermentation was extended after 48 h and no more sugar was consumed.

**Table 7.** Growth of *K. oxytoca* FMCC-197 under aerobic conditions in shake flasks using various initial sucrose concentrations employed. Representation of the maximum specific growth rate ($\mu_{max}$) of the strain, maximum biomass production ($X_{max}$), bioconversion yield of biomass produced per substrate consumed ($Y_{X/S}$), shake flask fermentations carried out at T = 30 °C. Culture conditions: growth in 500-mL flasks filled with 100 mL; agitation rate = 180 rpm; initial pH = 7.0. Each point is the mean value of two independent measurements.

| Initial Sucrose Concentration (g L$^{-1}$) | $\mu_{max}$ (h$^{-1}$) | Sugar Consumed (g L$^{-1}$) | $X_{max}$ (g L$^{-1}$) | $Y_{X/S}$ (g g$^{-1}$) | BDO (g L$^{-1}$) | Lag Phase Duration (h) | Fermentation Time (h) |
|---|---|---|---|---|---|---|---|
| ≈5 | 0.80 | 4.5 | 2.0 | 0.44 | 2.1 | 0.5 | 3.5 |
| ≈10 | 0.82 | 9.7 | 3.8 | 0.39 | 4.5 | 0.5 | 4.5 |
| ≈15 | 0.84 | 16.2 | 4.2 | 0.26 | 6.8 | 1.0 | 6 |
| ≈20 | 0.85 | 19.3 | 4.5 | 0.23 | 8.0 | 1.5 | 6.5 |
| ≈40 | 0.65 | 42.0 | 6.4 | 0.15 | 18.1 | 1.5 | 8 |
| ≈60 | 0.32 | 62.3 | 6.5 | 0.10 | 28.2 | 1.5 | 24 |
| ≈80 | 0.27 | 78.4 | 7.0 | 0.09 | 37.0 | 2.0 | 28 |
| ≈90 | 0.25 | 91.8 | 7.5 | 0.08 | 39.6 | 2.3 | 29 |
| ≈110 | 0.14 | 87.2 * | 6.2 | 0.07 | 38.8 | 2.5 | 32 * |
| ≈120 | 0.13 | 100.5 ** | 6.3 | 0.06 | 35.7 | 3.0 | 35 ** |
| ≈130 | 0.12 | 90.2 *** | 5.8 | 0.06 | 36.2 | 3.5 | 36 *** |
| ≈150 | 0.10 | 98.5 **** | 5.5 | 0.06 | 41.0 | 5.0 | 40 **** |
| ≈160 | 0.08 | 101.5 ***** | 6.0 | 0.06 | 38.0 | 7.0 | 50 ***** |

* Initial sugars at 110.4 g L$^{-1}$, fermentation was extended after 32 h and no more sugar was consumed; ** initial sugars at 122.1 g L$^{-1}$, fermentation was extended after 35 h and no more sugar was consumed; *** initial sugars at 133.2 g L$^{-1}$, fermentation was extended after 36 h and no more sugar was consumed; **** initial sugars at 148.2 g L$^{-1}$, fermentation was extended after 40 h and no more sugar was consumed; ***** initial sugars at 164.5 g L$^{-1}$, fermentation was extended after 50 h and no more sugar was consumed.

In the case of aerobic fermentations, high values of the specific growth rate ($\mu_{max} > 0.65$ h$^{-1}$) were obtained when the initial substrate concentration was lower than 40 g L$^{-1}$. When higher initial sucrose concentrations were employed, $\mu_{max}$ values decreased, suggesting that substrate inhibition is observed due to the relatively increased initial sucrose concentration added into the medium. It is worth mentioning that irrespec-

tive of the (sufficiently high in several trials) concentration of sucrose found in the medium, satisfactory BDO quantities (up to 41.0 g L$^{-1}$; simultaneous yield $Y_{BDO} \approx 0.41$–0.44 g g$^{-1}$) were recorded. On the other hand, the yield of biomass produced per substrate consumed ($Y_{X/S}$) constantly decreased with the whole range of sucrose concentrations employed, while the lag phase duration slightly increased with the increment of sucrose quantity in the medium, demonstrating the inhibition exerted by the increasing initial sucrose quantities towards the microbial growth. Given that *K. oxytoca* FMCC-197 was unable to assimilate sucrose quantities > 100 g L$^{-1}$ in batch experiments, fed-batch strategies followed to maximize BDO production.

It should be stressed that during the previous fermentations there was no pH regulation. The initial value was 7.0 ± 0.2 and remained uncontrolled until the end of the experiment. In the cases of 30 g L$^{-1}$ and 60 g L$^{-1}$ for the initial substrate concentrations, the lowest value noted was ≈6.0 at the end of the process. In higher initial values of carbon source concentration, the pH value at the end of the fermentation was ≈5.5.

Fed-batch experiments in flasks and bioreactors. To enhance the final BDO production, six fed-batch experiments were conducted under different conditions. Four fed-batch experiments were performed in a bioreactor using sugarcane molasses as the initial carbon source, with sucrose being added into the culture medium with intermittent feeding when needed. Two fed-batch experiments were performed in shake flasks using sucrose as the sole carbon source, with a previously sterilized or pasteurized medium. As shown in Table 8, the final BDO production was remarkably favored when aeration was applied into the fed-batch cultures at T = 30 °C. When molasses and sucrose were the fermentation substrate, 38 g L$^{-1}$ of BDO accumulated in 48 h of anaerobic cultivation (Figure 1) while the very high concentration of 115.3 g L$^{-1}$ of BDO was recorded within 64 h when aeration was applied throughout the whole fermentation (Table 8 and Figure 2). In addition, a remarkable increase in yield was observed under aerobic conditions reaching a value of 0.40 g g$^{-1}$, when under anaerobic conditions the yield was 0.34 g g$^{-1}$. Relevant observations were obtained concerning the productivity rate, as in the case of anaerobic conditions the value was 0.79 g L$^{-1}$ h$^{-1}$ and under aerobic conditions where the value was 1.80 g L$^{-1}$ h$^{-1}$. When the same aerobic fed-batch experiment was conducted at T = 37 °C, no significant difference in the fermentation duration was noted, however, the final BDO production was remarkably decreased, reaching a value of 71.8 g L$^{-1}$, accompanied by a decrease in the productivity rate. When molasses was employed as the sole carbon source at T = 30 °C, and pulses with concentrated molasses-based compounds were added into the culture medium when needed, both the yield and the productivity rate were reduced (Figure 3). This could be explained by the fact that molasses wastewater contained mainly melanoidins, and other compounds that inhibit bacterial growth when they are accumulated in the culture medium. As for the shake flask experiments, when the culture medium was previously sterilized, 145.4 g L$^{-1}$ of total sugars were consumed within 58 h, leading to the production of 60 g L$^{-1}$ of BDO. A remarkable yield of 0.41 g g$^{-1}$ was achieved, while the productivity rate was 1.03 g L$^{-1}$ h$^{-1}$. In the case of a previously pasteurized medium, 164.8 g L$^{-1}$ of substrate was converted into 58.0 g L$^{-1}$ of BDO. The bioconversion yield and productivity rate of the process were reduced as the values of 0.35 g g$^{-1}$ and 0.94 g L$^{-1}$ h$^{-1}$ were respectively noted. Nevertheless, the accomplishment of the fermentation with satisfactory results using not previously sterilized media demonstrates the potential of the strain *K. oxytoca* FMCC-197 to convert sucrose (and thus, sucrose-containing wastewaters and residues) into BDO in larger-scale operations, eliminating the cost of sterilization. The fed-batch profiles of BDO production along with growth and substrate consumption under anaerobic and aerobic conditions, as well as the fermentation using molasses as a sole carbon source are shown in Figures 1–3, respectively.

**Table 8.** Growth parameters, substrate consumption, final BDO and other organic acid concentrations, conversion yield ($Y_{BDO}$), and productivity ($P_{BDO}$) during fed-batch fermentations in a 2-L bioreactor, using molasses and sucrose as carbon sources under anaerobic and aerobic conditions. Culture conditions for the bioreactor experiments: T = 30 °C or 37 °C; aeration rate = 1 vvm (aerobic conditions); agitation rate: from 180 rpm to 400 rpm (aerobic conditions); agitation rate 180 rpm (anaerobic conditions); pH fluctuating from 7.0 to 6.0. Culture conditions for the shake flask experiments: growth in 2-L flasks filled with 500 mL, T = 30 °C; agitation rate: 180 rpm; pH fluctuating from 7.0 to 6.0. Each point is the mean value of two independent measurements.

| Fermentation Mode | Aeration Mode | T °C | Total Sugars Consumed (g L$^{-1}$) | X$_{max}$ (g L$^{-1}$) | BDO (g L$^{-1}$) | Acetoin (g L$^{-1}$) | Ethanol (g L$^{-1}$) | Succinic (g L$^{-1}$) | Lactic (g L$^{-1}$) | Y$_{BDO}$ (g g$^{-1}$) | P$_{BDO}$ (g L$^{-1}$ h$^{-1}$) | Fermentation Time * (h) |
|---|---|---|---|---|---|---|---|---|---|---|---|---|
| Bioreactor | Anaerobic [a] | 30 | 110.1 ± 2.0 | 6.6 ± 1.4 | 38.0 ± 1.7 | 4.2 ± 1.0 | 5.0 ± 1.6 | 3.5 ± 0.5 | 5.0 ± 1.2 | 0.34 ± 0.01 | 0.79 ± 0.01 | 48 |
|  | Aerobic [a] | 30 | 288.8 ± 2.0 | 9.0 ± 1.5 | 115.3 ± 1.8 | 5.2 ± 1.0 | 7.5 ± 1.5 | 5.5 ± 1.5 | 8.0 ± 1.5 | 0.40 ± 0.02 | 1.80 ± 0.02 | 64 |
|  | Aerobic [b] | 30 | 112.8 ± 1.8 | 7.4 ± 1.9 | 43.1 ± 1.8 | 3.5 ± 1.5 | 4.0 ± 0.5 | 4.5 ± 1.0 | 7.0 ± 1.5 | 0.38 ± 0.01 | 0.86 ± 0.01 | 50 |
|  | Aerobic [a] | 37 | 178.2 ± 3.0 | 7.9 ± 2.1 | 71.8 ± 2.0 | 3.0 ± 1.0 | 4.2 ± 1.0 | 4.5 ± 1.1 | 4.6 ± 1.6 | 0.40 ± 0.01 | 1.09 ± 0.02 | 66 |
| Shake flask | Aerobic [c] | 30 | 145.4 ± 4.0 | 9.5 ± 1.5 | 60.0 ± 2.0 | 3.5 ± 1.2 | 4.5 ± 1.0 | 4.0 ± 0.5 | 4.0 ± 0.2 | 0.41 ± 0.01 | 1.03 ± 0.03 | 58 |
|  | Aerobic [d] | 30 | 164.8 ± 5.1 | 13.2 ± 2.5 | 58.0 ± 1.5 | 4.0 ± 0.5 | 3.8 ± 0.2 | 5.1 ± 1.1 | 5.5 ± 1.5 | 0.35 ± 0.02 | 0.94 ± 0.01 | 62 |

* Fermentations were extended after the indicated time and no more sugar was consumed. [a]: Molasses and sucrose as a carbon source; [b]: molasses as the sole carbon source; [c]: sucrose as the sole carbon source; previously sterilized medium; [d]: sucrose as the sole carbon source; previously pasteurized medium.

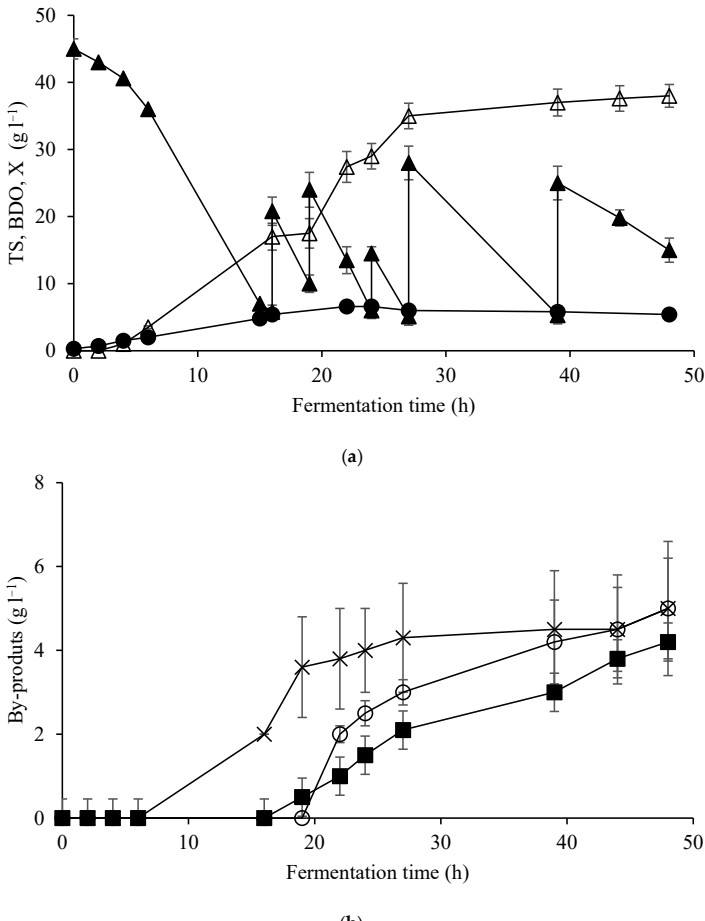

(**a**)

(**b**)

**Figure 1.** Kinetics of evolution of (**a**) total sugars (TS, g L$^{-1}$) (▲), 2,3-butanediol (BDO, g L$^{-1}$) (△), biomass (X, g L$^{-1}$) (●); (**b**) acetoin (Ace, g L$^{-1}$) (■), lactic acid (Lac, g L$^{-1}$) (○), and ethanol (Eth, g L$^{-1}$) (×) during growth of *K. oxytoca* FMCC-197 on molasses and pulses of sucrose in fed-batch bioreactor experiments. Culture conditions: anaerobic trial, 180 rpm agitation rate, pH fluctuating from 7.0 to 6.0, T = 30 °C, growth in 2-L bioreactor. Each point is the mean value of two independent measurements.

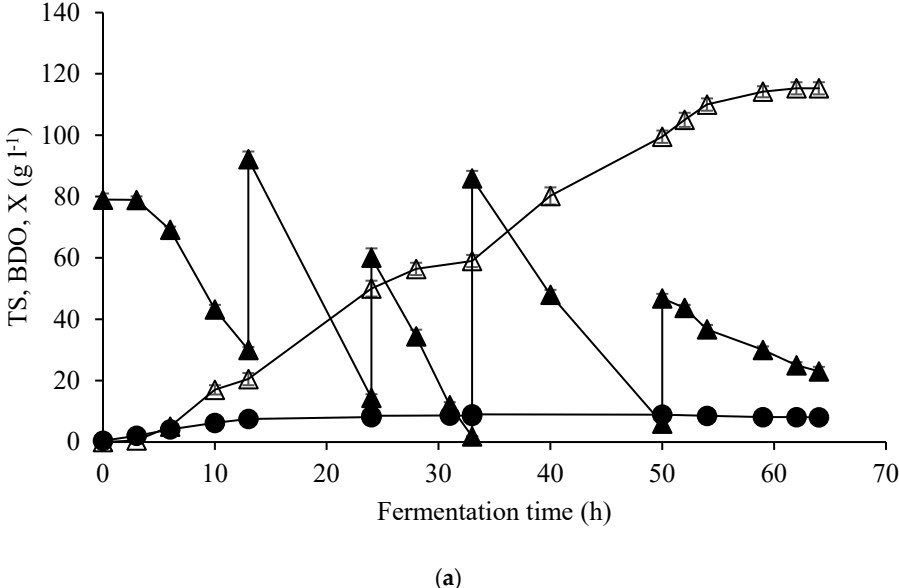

(**a**)

**Figure 2.** *Cont.*

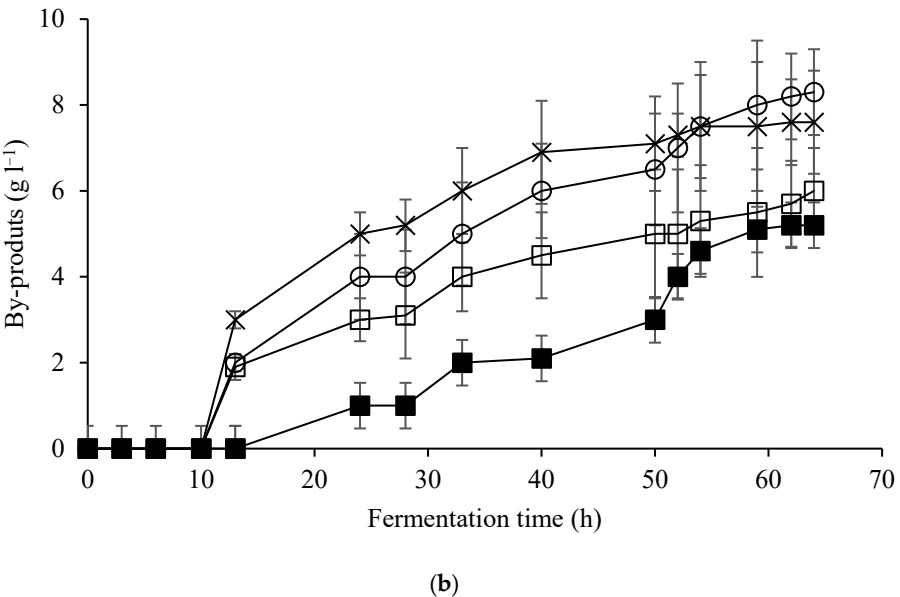

(**b**)

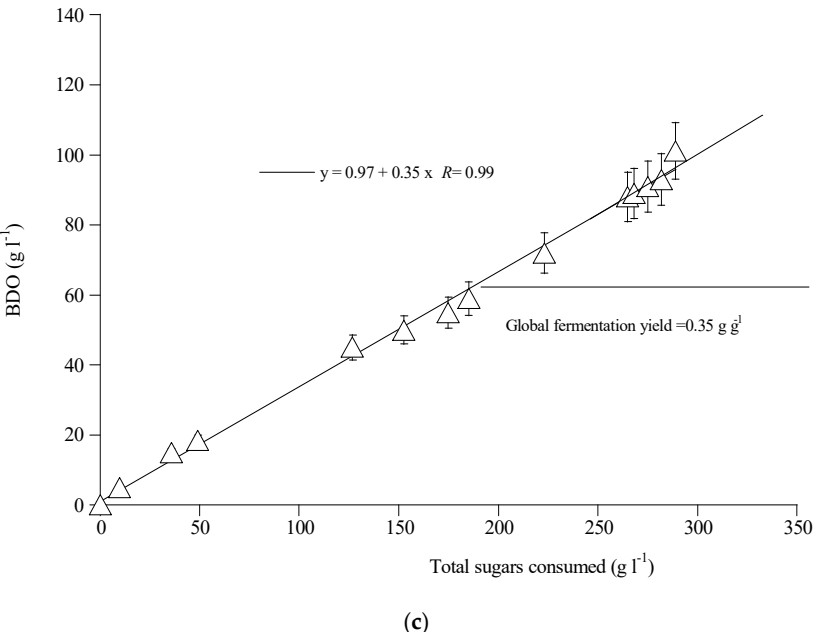

(**c**)

**Figure 2.** Kinetics of evolution of (**a**) total sugars (TS, g L$^{-1}$) (▲), 2,3-butanediol (BDO, g L$^{-1}$) (△), biomass (X, g L$^{-1}$) (●); (**b**) acetoin (Ace, g L$^{-1}$) (■), lactic acid (Lac, g L$^{-1}$) (○) ethanol (Eth, g L$^{-1}$) (×), and succinic acid (Suc, g L$^{-1}$) (□); and (**c**) representation of 2,3-butanediol (BDO, g L$^{-1}$) (△) per sugar consumed, during growth of *K. oxytoca* FMCC-197 on molasses and pulses of sucrose in fed-batch bioreactor experiments. Culture conditions: 1 vvm aeration, agitation rate from 180 to 400 rpm, pH fluctuating from 7.0 to 6.0, T = 30 °C, growth in 2-L bioreactor. Each point is the mean value of two independent measurements.

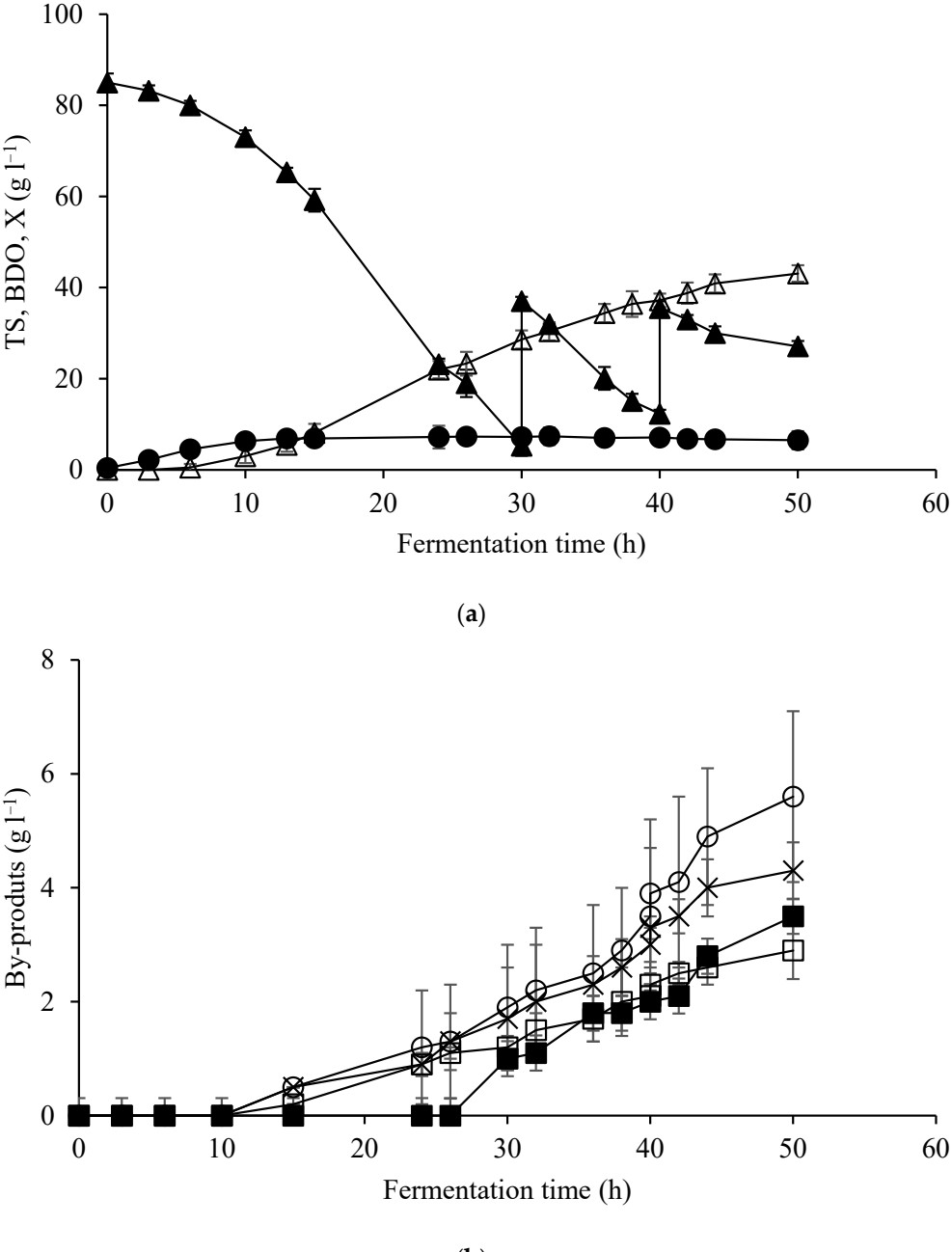

(**a**)

(**b**)

**Figure 3.** Kinetics of evolution of (**a**) total sugars (TS, g $L^{-1}$) (▲), 2,3-butanediol (BDO, g $L^{-1}$) (△), biomass (X, g $L^{-1}$) (●); (**b**) acetoin (Ace, g $L^{-1}$) (■), lactic acid (Lac, g $L^{-1}$) (○), ethanol (Eth, g $L^{-1}$) (×), and succinic acid (Suc, g $L^{-1}$) (□) during growth of *K. oxytoca* FMCC-197 on molasses in fed-batch bioreactor experiments. Culture conditions: 1 vvm aeration, agitation rate from 180 to 400 rpm, pH fluctuating from 7.0 to 6.0, T = 30 °C, growth in 2-L bioreactor. Each point is the mean value of two independent measurements.

Color removal occurring during molasses fermentation. Besides BDO production, the decolorization of molasses in trials where it was used as the sole carbon source was assessed. The anaerobic cultures were accompanied by a decolorization of the medium of *c.* 40%. On the other hand, during aerobic fermentations on molasses, a higher decolorization of the medium (≈50%) was achieved. In both aerobic and anaerobic experiments performed, decolorization seemed to be a completely growth-associated process. Figure 4 represents the color removal in both cases.

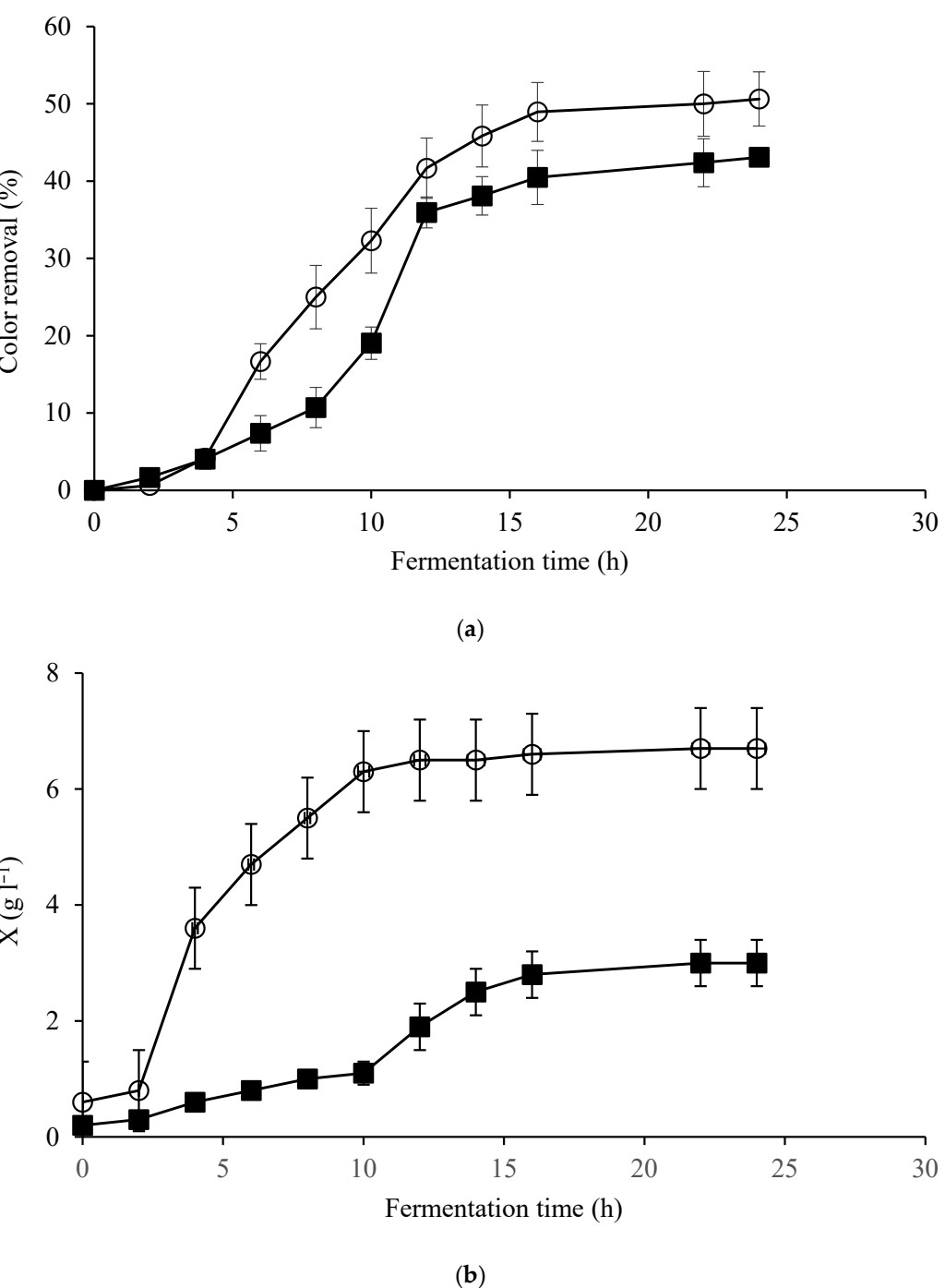

**(a)**

**(b)**

**Figure 4.** (**a**) Color removal during growth of *K. oxytoca* FMCC-197 on molasses under anaerobic (■) and aerobic (○) conditions, and (**b**) growth of *K. oxytoca* FMCC-197 on molasses under anaerobic (■) and aerobic (○) conditions. Culture conditions: T = 30 °C, 180 rpm agitation rate, initial pH = 7.0, initial total sugars concentration ≈ 30 g $L^{-1}$. Anaerobic experiments were conducted in Duran bottles and aerobic experiments on 500-mL shake flasks. Each point is the mean value of two independent measurements.

Effect of dissolved oxygen concentration upon the BDO production. Shake flask trials using three different carbon sources (*viz.* glucose, fructose, and sucrose) at an initial concentration of 30 g $L^{-1}$ were performed with the aim of studying the effect of the oxygen concentration and the oxygen uptake rate. Figure 5 presents the dissolved oxygen tension (DOT) and the specific oxygen consumption ($q_{O2}$) along with fermentation time and the related BDO production using the three carbon sources. As shown, DOT (%, *v/v*) evolution

as a function of the fermentation time demonstrated no significant differences during the fermentations; even at the early growth steps of the culture, DOT values decreased. The lowest value was noted after 18 h of fermentation, and it was $\approx 45\%$ $v/v$ in the case of glucose and fructose. In fermentations where sucrose was applied, the lowest DOT value recorded was $\approx 60\%$ $v/v$ after 12 h. Additionally, the representation of the specific consumption rate of oxygen for all trials demonstrates a significant respiratory activity for the strain at the first growth steps ($q_{O2} = 0.40 \pm 0.05$ g (g h)$^{-1}$) that remarkably decreased at the late fermentation steps of the culture ($q_{O2} \approx 0.1$ g (g h)$^{-1}$). These findings indicate that the biosynthesis of BDO in *K. oxytoca* FMCC-197 is achieved regardless of the physiological state of the culture regarding its respiratory activity. Following this, DOT and $q_{O2}$ values were determined for four different initial sucrose concentrations added into the medium (*viz.* 30, 60, 90, and 150 g L$^{-1}$) and the results are illustrated in Figure 6. As shown, the more the initial concentration of sucrose increased, the lower the DOT value was as observed in the flasks as the fermentations proceeded. However, it must be pointed out that in almost all trials, DOT concentrations remained almost always in values > 20–25% $v/v$, providing the growth environment with fully aerobic conditions [42] for the period of biosynthesis and accumulation of BDO. In the late fermentation steps and in the trial in which a very high initial sucrose concentration had been added into the medium ($\approx 150$ g L$^{-1}$), DOT values $\approx 10\%$ $v/v$ were detected, suggesting that the biosynthesis of BDO under aerobic/micro-aerobic conditions [41,42] occurred for the given period in the trial performed. The monitoring of $q_{O2}$ evolution during the fermentation demonstrated significant respiratory activity for the strain at the first growth steps ($q_{O2}$ ranging between 0.3 and 0.6 g (g h)$^{-1}$) that decreased as the fermentation proceeded (specifically, in the trials with initial sucrose concentrations adjusted to 90 and 150 g L$^{-1}$, the respiratory activity, as correlated with the achieved $q_{O2}$ values, was indeed very low). From the results obtained, it is suggested that sufficient BDO production was performed under aerobic conditions, although this is not a common feature for the bacterial metabolism, since the BDO bioconversion process is mostly considered as an "anaerobic" or "micro-aerobic" one [7,8,12].

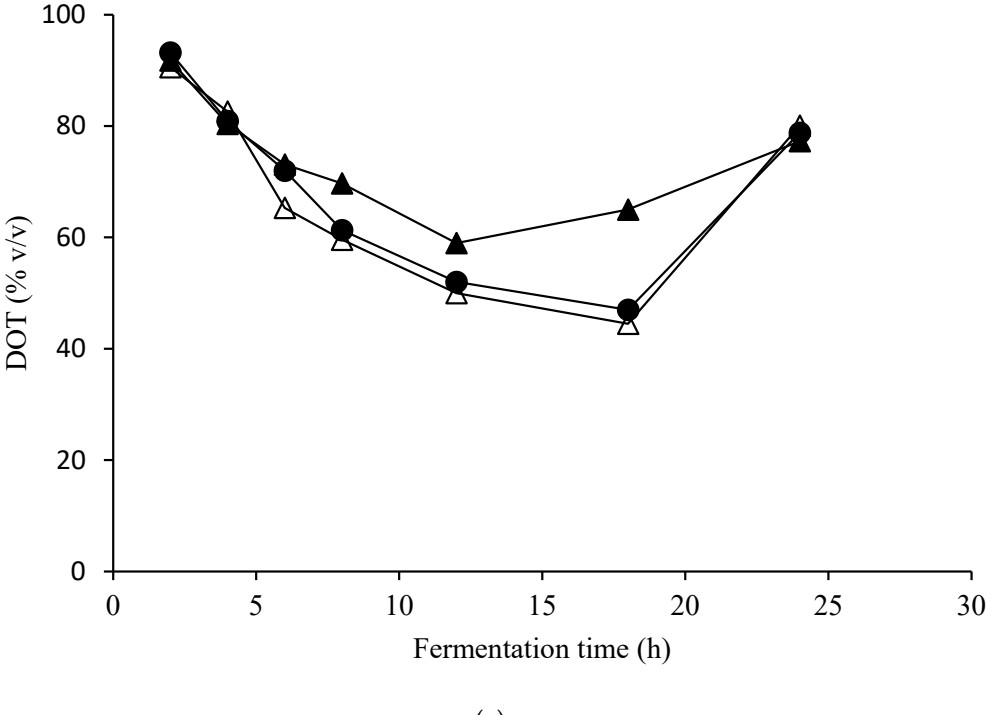

(**a**)

**Figure 5.** *Cont.*

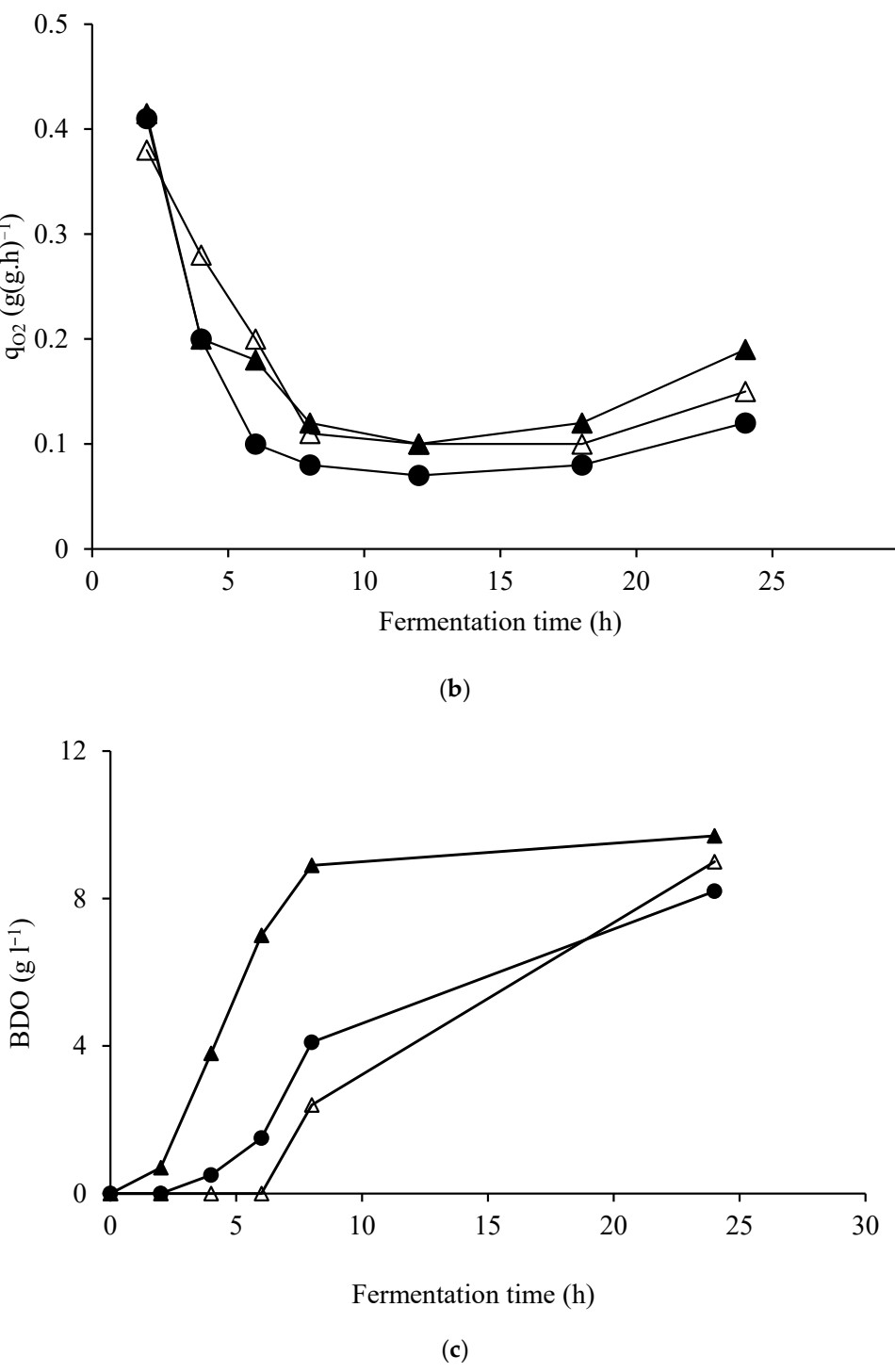

(**b**)

(**c**)

**Figure 5.** (**a**) Dissolved oxygen tension (DOT, % *v/v*), (**b**) specific oxygen consumption rate ($q_{O2}$, g (g h)$^{-1}$), and (**c**) 2,3-butanediol (BDO, g L$^{-1}$) production related with fermentation time during shake flask cultivation of *K. oxytoca* FMCC-197 on glucose (•), fructose (△), and sucrose (▲). Culture conditions: T = 30 °C, 180 rpm agitation rate, initial pH = 7.0, initial glucose concentration ≈30 g L$^{-1}$.

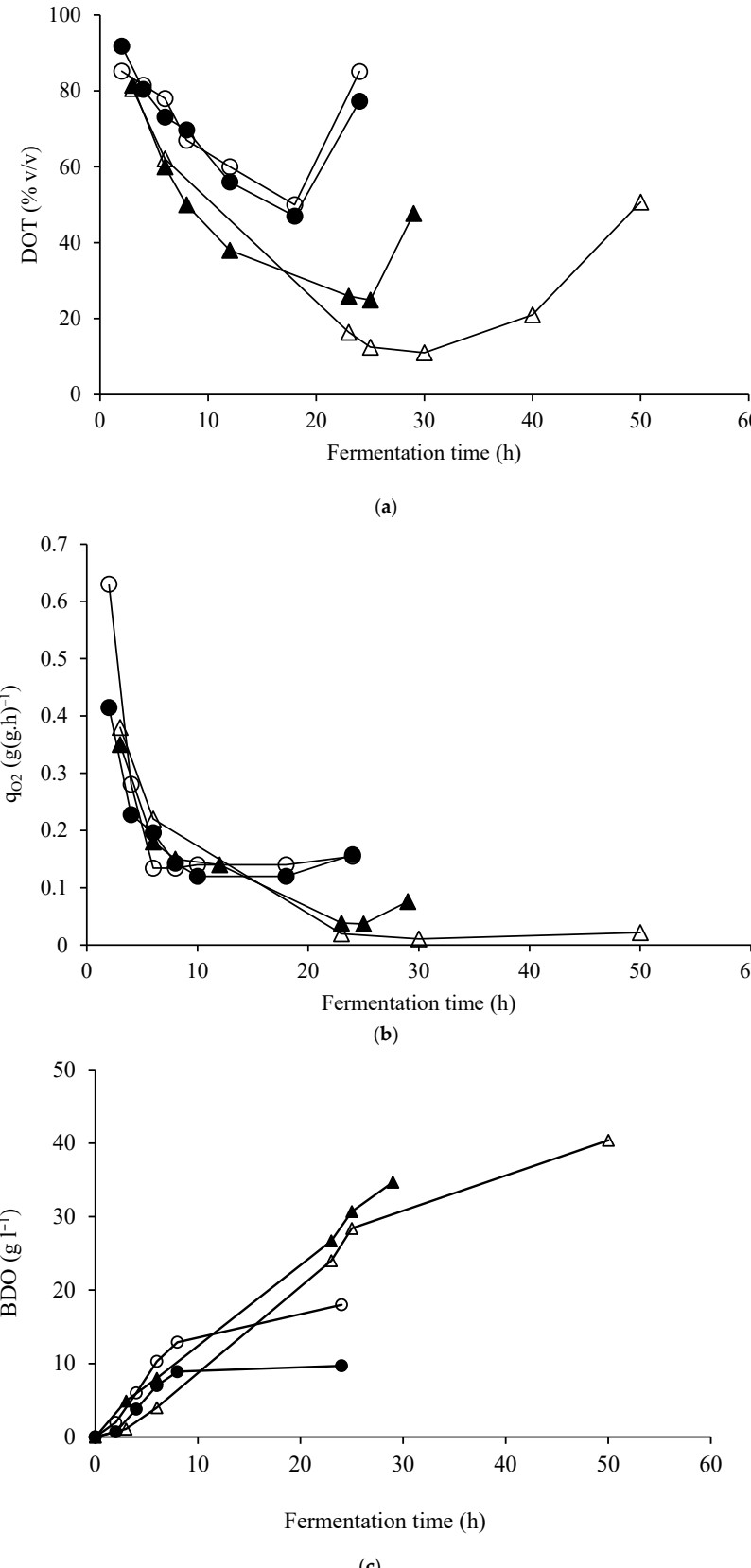

**Figure 6.** (**a**) Dissolved oxygen tension (DOT, % *v/v*), (**b**) specific oxygen consumption rate ($q_{O2}$, g (g h)$^{-1}$), and (**c**) 2,3-butanediol (BDO, g L$^{-1}$) production related to fermentation time during shake flask cultivation of *K. oxytoca* FMCC-197 on four different initial sucrose concentrations: 30 (●), 60 (○), 90 (▲) and 150 (△) g L$^{-1}$. Culture conditions: T = 30 °C, 180 rpm agitation rate, initial pH = 7.0.

## 4. Discussion

Waste streams from various industrial activities can be used as renewable substrates for the microbiological production of metabolites of high biotechnological interest. One such metabolite is BDO, which can be produced via microbial bioconversion using predominantly bacterial strains, although recent research has shown that yeasts and marine microalgae can also be used as BDO producers [12,14,17,43,44]. BDO and its derivatives are platform chemicals with a wide range of applications in many fields.

In this study, various newly isolated strains were assessed for their potential to convert sucrose into BDO during initial anaerobic cultivations in Duran bottles. Among them, the strain *K. oxytoca* FMCC-197 was proven to be the optimal one with significant yield and productivity rates and it was selected for further investigation to determine the optimum cultural conditions for an economical bioprocess. From the results obtained during anaerobic and aerobic conditions, it was shown that BDO production was significantly enhanced in the presence of $O_2$, leading to ameliorated productivity rates. Considering that aeration is the crucial factor for high biomass production and BDO synthesis, the following experiments were conducted under aerobic conditions. The strain could grow even in high carbon source concentrations without substrate inhibition. The productivity rate was dramatically decreased when over 90 g $L^{-1}$ of the sucrose concentration was initially added into the media, while the conversion yield was still satisfying.

The renewable resources that can be used for efficient and environmentally friendly bioprocesses can be either non-cellulosic or lignocellulosic. Considering that food wastes, molasses, whey, starch hydrolysates from cereals, and crude glycerol belong to the noncellulosic substrates and consist of various monosaccharides and polysaccharides, the strain *K. oxytoca* FMCC-197 was applied in aerobic cultivations using alternative carbon sources. The potential of the strain to convert different hexoses and pentoses into BDO proves that it can be efficiently used in the above low-cost residues, which only require supplementation with small quantities of mineral salts and nitrogen sources. Current research supports several studies that propose that the conversion of food waste into BDO via microbial fermentation is a promising way to reduce waste disposal into the environment and produce sustainable chemicals [45–47]. Additionally, the strain *K. oxytoca* FMCC-197 could ferment xylose, which encourages further investigation into using lignocellulosic biomass such as press cake from fruits and vegetables as a substrate that can lead to satisfying BDO production. The highest productivity rate that corresponds to a more economical bioprocess was at 30 °C, although the strain could grow in a wide range of temperature values. Focusing on the optimization of the process to maximize the final BDO production, fed-batch fermentations were conducted in scaled-up bioreactor cultivations using different substrates and aeration strategies.

From the results obtained during the fed-batch experiments, the maximum final BDO concentration reached a value of 115.3 g $L^{-1}$ at around 64 h, which corresponds to a remarkable final productivity rate of 1.8 g $L^{-1}$ $h^{-1}$. The strain revealed a satisfying ability to grow on pasteurized media, surpassing the biomass of other strains and produced 48 g $L^{-1}$ of BDO, reaching a yield of 0.35 g $L^{-1}$. In addition, the capability of the strain to decolorize molasses (>40%) through melanoidins breakdown or adsorption into the cell surface, under both anaerobic and aerobic conditions, suggests that molasses wastewaters derived after BDO fermentation by *K. oxytoca* FMCC-197 can be disposed relatively safely to the environment. These findings offer a novel perspective on the issue of molasses effective disposal, presenting an innovative approach of bacterial fermentation instead of classical techniques using ultrafiltration or enzymes [48]. Finally, according to the results obtained from dissolved oxygen tension (DOT) and the specific oxygen consumption ($q_{O2}$) assessment, it may be assumed that in all cases the fermentations were conducted under full aerobic conditions [42,49]. The values noted are considered to be the lower thresholds of concentrations for aerobic metabolism in several types of yeasts, fungi, and bacteria [40,42,50,51]. Therefore, from all the above-mentioned analysis, it can be stated that although BDO fermentation is mostly considered as an "anaerobic" or "micro-aerobic"

bioprocess [7,8], the strain *K. oxytoca* FMCC-197 revealed great potential for BDO production under aerobic conditions.

The results obtained throughout this study revealed the ability of the newly isolated strain *K. oxytoca* FMCC-197 to be used as a natural BDO producer via the fermentation route. This wild strain is competitive against various wild and genetically modified bacteria, which belong mainly to the genus *Klebsiella*. A synopsis of the results reported in the literature and their comparison with the current investigation concerning BDO production using commercial and low-cost substrates is presented in Table 9. The results reveal that this novel strain can be a competitive BDO producer compared to other strains in the literature that have also demonstrated remarkable production. For instance, the main advantage of the strain *K. oxytoca* FMCC-197 is that it is a newly isolated, wild-type, and food-derived one, which can be used directly in the biotechnological production of BDO, reaching high yields and productivity rates comparable to those obtained using genetically modified strains. As mentioned, in the literature, many other competitive strains that present similar results with the present study (i.e., strains ME-UD-3, PDL-K5, etc.) are genetically modified ones.

Among other wild-type strains with similar results is the strain *Enterobacter ludwigii*, which has demonstrated a remarkable ability to convert molasses to BDO during an optimized fed-batch fermentation leading to the production of 50.6 g $L^{-1}$ BDO with a productivity rate of 2.66 g $L^{-1}$ $h^{-1}$ [52]. Recently, the strain *Bacillus amyloliquefaciens* was able to ferment molasses and produce mainly the (2R,3R)-BDO isomer during a fed-batch fermentation with a yield of 0.40 g $g^{-1}$ and productivity of 0.83 g $L^{-1}$ $h^{-1}$ [53].

**Table 9.** BDO production by different bacterial strains of the *Klebsiella* genus.

| Strain | Substrate | BDO (g $L^{-1}$) | Yield (g $g^{-1}$) | Fermentation Mode | Reference |
|---|---|---|---|---|---|
| *K. oxytoca* ME-UD-3 | Glucose | 130 | 0.48 | Fed-batch/Bioreactor | [3] |
| *K. pneumoniae* PM2 | Parm tree hydrolysate | 75.03 | 0.43 | Fed-batch/Bioreactor | [54] |
| *K. oxytoca* PDL-K5 | Whey powder | 74.9 | 0.43 | Fed-batch/Bioreactor | [55] |
| *K. oxytoca* CHA006 | Lignocellulosic hydrolysates | 6.11 | 0.34 | Batch | [56] |
| *K. oxytoca* M1 | Glucose | 19.0 | 0.32 | Batch/ Shake flasks | [57] |
| | Xylose | 17.1 | 0.28 | | |
| | Galactose | 15.1 | 0.25 | | |
| | Fructose | 18.2 | 0.29 | | |
| *K. oxytoca* NBRF4 | Glucose | 34.2 | 0.35 | Fed-batch/Bioreactor | [58] |
| *K. oxytoca* M1 | Glucose | 118.5 (+42.1 Ace) | 0.46 | Fed-batch/Bioreactor | [59] |
| *K. oxytoca* M3 | Crude glycerol | 131.5 | 0.44 | Fed-batch/Bioreactor | [60] |
| *K. pneumoniae* CICC 10011 | Glucose | 52.4 | 0.38 | Batch/ Shake flasks | [61] |
| *K. pneumoniae* SDM | Corn steep liquor | 151 | 0.48 | Fed-batch/Bioreactor | [15] |
| *K. pneumoniae* SDM | Corncob molasses | 78.9 | 0.41 | Fed-batch/Bioreactor | [62] |
| *K. pneumoniae* CGMCC 1.9131 | Glucose | 20.93 | 0.38 | Batch/ Shake flasks | [63] |
| | Xylose | 11.1 | 0.49 | | |
| | Sugarcane acid hydrolysate | 17.35 | 0.43 | | |
| | Sugarcane alkali hydrolysate | 14.53 | 0.43 | | |
| *K. pneumoniae* G31 | Glycerol | 70 | 0.39 | Fed-batch/Bioreactor | [64] |

**Table 9.** *Cont.*

| Strain | Substrate | BDO (g L$^{-1}$) | Yield (g g$^{-1}$) | Fermentation Mode | Reference |
|---|---|---|---|---|---|
| *K. pneumoniae* CICC 10781 | Cheese whey powder | 57.6 | 0.40 | Fed-batch/ Bioreactor | [65] |
| *K. oxytoca* ACA-DC 1581 | Biodiesel-derived glycerol | 69.0 | 0.47 | Fed-batch/ Bioreactor | [11] |
| *Klebsiella oxytoca* FMCC-197 | Molasses + Sucrose<br>Molasses + Sucrose<br>Sucrose | 115.3<br>71.8<br>60.0 | 0.40<br>0.40<br>0.41 | Fed-batch/Bioreactor<br>Fed-batch/Shake flask | Current study |

## 5. Conclusions

The present study demonstrated the potential of a newly isolated *K. oxytoca* strain to produce significant BDO quantities under various fermentation configurations. BDO biosynthesis and secretion were influenced by several fermentation parameters, such as the nature and the initial concentration of the carbon source used, the incubation temperature, and the imposed aeration. Considering the results of the present research, the development of an efficient and economical profitable plan, including the cost of manufacturing as well as fixed capital investment [4], using the strain *K. oxytoca* FMCC-197, can lead to the production of BDO along with its derivatives at an industrial level.

**Author Contributions:** A.M.P.: investigation, methodology, data curation, writing—original draft, visualization; E.I.M.D.: investigation, methodology; A.A.K.: conceptualization, writing—review and editing, supervision, funding; S.P.: conceptualization, writing—original draft, writing—review and editing, supervision, project administration. All authors have read and agreed to the published version of the manuscript.

**Funding:** This research received no external funding.

**Institutional Review Board Statement:** Not applicable.

**Informed Consent Statement:** Not applicable.

**Data Availability Statement:** Data are unavailable due to privacy restrictions. Researchers can provide if necessary.

**Conflicts of Interest:** The authors declare no conflict of interest.

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
