# Peer review of "Screening Bacterial Strains Capable of Producing 2,3-Butanediol: Process Optimization and High Diol Production by Klebsiella oxytoca FMCC-197"

_fermentation, doi:10.3390/fermentation9121014_

Round 1
Reviewer 1 Report
Comments and Suggestions for Authors
This manuscript explored screening bacterial strains capable of producing 2,3-butanediol: process optimization and high diol production by Klebsiella oxytoca FMCC-197. The theme is interesting and worthy of research. However, the manuscript needs some revision before acceptance for publication.
Abstract
Too long. Please condense it. Meanwhile, please add some quantitative data and highlight the the innovation of this study.
Introduction
Please Please make appropriate modifications. A good introduction should conclude the introduction by mentioning the specific objectives of the research and the earlier paragraphs should lead logically to specific objectives of the study.
-Revised Introduction section based on the structure below:
1st paragraph: Problem statement
2nd paragraph: Current ongoing solution
3rd paragraph: Proposed solution in this work.
4th paragraph: Summarized the current research novelty and objective of this work.
Materials and Methods
Line 115-120. How are strains screened? Please briefly describe the screening process. Why is Duran bottle fermentations? Not the serum bottles. What is the difference between the Fed-batch bioreactor fermentations and Fed-batch fermentations in shake-flasks.
Results
Table 1-3. What are the YBDO and PBDO? How do they calculate?
Table 4. Why is the fermentation time inconsistent?
Line 308-309. What is the basis for temperature setting?
Table 6. How to set the initial sucrose concentration and fermentation time? Initial sucrose concentration looks too close. Similar to Table 7.
The author has explored the impact of different influencing factors on the production of 2,3-butanediol. What is the final conclusion drawn? For example, the optimal fermentation conditions. At the same time, which factor is the most important? Please clarify it.
Discussion
The discussion section is not in-depth enough, please strengthen it.
Suggest providing a “5. conclusion”.
Reference
Some references are too old, please cite the latest references.
There are still some grammar and formatting errors in the manuscript. Please check the entire text carefully and correct the grammar and formatting errors.
Comments on the Quality of English LanguageThere are still some grammar and formatting errors in the manuscript. Please check the entire text carefully and correct the grammar and formatting errors.
Reviewer 2 Report
Comments and Suggestions for Authors
- Table 5 show results for aerobic conditions, but typical anaerobic metabolites are obtained (ethanol and lactic acid) although in low amounts. Please explain.
- The aerobic (or anerobic) condition should be specify in label of Table 7.
- Results presented are complete and very detailed. This leads to a loss of focus during reading and comprehending each approach. Too much numerical information in tables and figures should be summarized in a single (or double) table for best results under aerobic and anaerobic conditions corresponding to batch fermentations.
- Discussion should elaborate on quantitative result based on tables and graphs previously presented. Avoid repeating numerical figures.
Comments on the Quality of English LanguageEnglish is acceptable with minor revision needed.
Reviewer 3 Report
Comments and Suggestions for Authors
Line 323, Table 5, the difference in the operating temperature measurement is varying from 2-3.
i. During industrial operations it is very difficult to maintain this small temperature difference.
ii. Did you evaluate the accuracy of the temperature measuring device in your lab work?
Line 461, Table 8, Culture conditions for the bioreactor experiments: T=30 °C or 37 °C. Please discuss why for same type of carbon source, increase in temperature sugar consumption was increased but the fermentation time given was less i.e. 55 hr at 30 OC and 66 hours at 37 OC …… This comparison cannot be appreciated when the parameters are not same.
Line 620, Table 9, The claim of the authors that their selected strain give significant BDO is not true because some of the strains enlisted here have more yield than Klebsiella oxytoca FMCC-197. Therefore, they need to rewrite discussion and abstract part where such things are mentioned.

English is OK but scientific presentation of table , figures is not appropriate
Round 2
Reviewer 1 Report
Comments and Suggestions for Authors
This manuscript can be accepted.
Author Response
I would like to express my heartfelt gratitude for your contribution as a reviewer and for the acceptance of our manuscript for publication. My research team and I sincerely appreciate your time, effort, and comments which played a pivotal role in the production of a high quality research manuscript.
Reviewer 3 Report
Comments and Suggestions for Authors
Looks like that you cannot do more experiments then you can justify the concerns in discussion, please.
Have not addressed the following in the second version of paper:
Line 323, Table 5, the difference in the operating temperature measurement is varying from 2-3.
i. During industrial operations it is very difficult to maintain this small temperature difference.
ii. Did you evaluate the accuracy of the temperature measuring device in your lab work?
Line 461, Table 8, Culture conditions for the bioreactor experiments: T=30 °C or 37 °C. Please discuss why for same type of carbon source, increase in temperature sugar consumption was increased but the fermentation time given was less i.e. 55 hr at 30 OC and 66 hours at 37 OC …… This comparison cannot be appreciated when the parameters are not same.
Line 620, Table 9, The claim of the authors that their selected strain give significant BDO is not true because some of the strains enlisted here have more yield than Klebsiella oxytoca FMCC-197. Therefore, they need to rewrite discussion and abstract part where such things are mentioned.
